# Mechanism-Based Fault Diagnosis Deep Learning Method for Permanent Magnet Synchronous Motor

**DOI:** 10.3390/s24196349

**Published:** 2024-09-30

**Authors:** Li Li, Shenghui Liao, Beiji Zou, Jiantao Liu

**Affiliations:** 1School of Automation, Central South University, Changsha 410083, China; lilicsu@csu.edu.cn; 2School of Computer Science and Engineering, Central South University, Changsha 410083, China; bjzou@csu.edu.cn; 3School of Mechanical Engineering and Automation, Fuzhou University, Fuzhou 350108, China

**Keywords:** convolutional neural network, continuous wavelet transform, fault diagnosis, mechanism analysis, permanent magnet synchronous motor

## Abstract

As an important driving device, the permanent magnet synchronous motor (PMSM) plays a critical role in modern industrial fields. Given the harsh working environment, research into accurate PMSM fault diagnosis methods is of practical significance. Time–frequency analysis captures the rich features of PMSM operating conditions, and convolutional neural networks (CNNs) offer excellent feature extraction capabilities. This study proposes an intelligent fault diagnosis method based on continuous wavelet transform (CWT) and CNNs. Initially, a mechanism analysis is conducted on the inter-turn short-circuit and demagnetization faults of PMSMs, identifying and displaying the key feature frequency range in a time–frequency format. Subsequently, a CNN model is developed to extract and classify these time–frequency images. The feature extraction and diagnosis results are visualized with t-distributed stochastic neighbor embedding (t-SNE). The results demonstrate that our method achieves an accuracy rate of over 98.6% for inter-turn short-circuit and demagnetization faults in PMSMs of various severities.

## 1. Introduction

The permanent magnet synchronous motor (PMSM) is extensively used in electric vehicles, rail transit, aerospace, and other fields, owing to its high power density, high efficiency, and broad speed regulation range [1,2]. Operating under harsh conditions such as high temperatures, high pressure, overcurrent, and overvoltage for extended periods, the motor is prone to failures and shutdowns that can potentially cause traffic accidents and substantial economic losses [3,4]. Common fault types in PMSMs primarily include stator winding short-circuit inter-turn faults, rotor eccentric faults, permanent magnet demagnetization faults, and motor bearing faults. Among these, inter-turn short-circuit and permanent magnet demagnetization faults are typical and particularly destructive. Inter-turn short-circuit faults account for 21% of all faults [5,6], while permanent magnet demagnetization faults represent 26% of all faults [7,8]. Therefore, diagnosing inter-turn short-circuit and permanent magnet demagnetization faults in PMSMs is of great value and significance. Fault diagnosis technology includes three components: fault signal acquisition, feature extraction, and pattern recognition [9]. For permanent magnet synchronous motors, fault signal acquisition primarily involves detecting current, voltage, torque, temperature, and magnetic flux. Feature extraction involves analyzing the stator current signal and other data collected from the motor to extract characteristic information representing the motor’s current operating state. Based on these two aspects, pattern recognition is then applied to classify this characteristic information to identify whether the motor is faulty and to determine the type and severity of the fault.

Currently, fault diagnosis methods for permanent magnet synchronous motors can be roughly categorized into three groups: those based on mathematical models, those relying on signal analysis, and those utilizing artificial intelligence [10]. The fault diagnosis method based on mathematical models offers high precision, provided that an accurate mathematical model is established. Vaseghi B. et al. developed a mathematical model for simulating inter-turn short-circuit faults in permanent magnet synchronous motors. They demonstrated that an increase in the number of short-circuited turns in the motor windings results in a more pronounced increase in the fault phase current and greater imbalance in the three-phase current [11,12]. Thierry B. et al. utilized the third harmonic of the stator current to analyze and identify motor faults using a mathematical model established for permanent magnet synchronous motors [13]. Although the fault diagnosis method based on mathematical models offers high precision, it requires the establishment of an accurate mathematical model. The calculations involved can be relatively complex, making it unsuitable for rapid diagnosis. In such cases, rapid fault diagnosis methods can replace extensive calculations. The fault diagnosis method based on signal processing involves detecting and collecting the output signals of the motor drive system, such as voltage and current. Its objective is to obtain the time domain and frequency domain characteristics of faults through various signal processing methods to diagnose the motor’s fault type. Jiang Z. et al. extracted the current signal of demagnetization faults using a simulation model and employed the wavelet transform (WT) signal processing method to extract the frequency domain characteristics of the back EMF for diagnosing demagnetization faults [14]. Dalahmeh M. et al. detected demagnetization faults by using FFT and HHT transformations to extract the frequency domain characteristics of the harmonic signal of the stator current [15]. Another study achieved the diagnosis and localization of inter-turn short-circuit faults by extracting the fundamental wave and the corresponding harmonic amplitude characteristics of the air-gap flux in inter-turn short-circuit faults [16]. While the fault diagnosis method based on signal processing does not require high accuracy of the model, which can save time, it demands high accuracy of the sensor and is susceptible to interference from external environmental factors, leading to significant deviations in the diagnosis results.

With the development and progress of computer technology and deep learning algorithms, artificial intelligence (AI) technology has been applied in various fields, such as the widespread use of Chat Generative Pretrained Transformer (ChatGPT) [17]. The fault diagnosis method based on artificial intelligence does not require the establishment of a complex mathematical model. Instead, it solely extracts and analyzes key information from the motor drive system, utilizing technologies like deep learning algorithms to detect the motor’s state and effectively judge motor faults. Kao et al. collected the stator current signal of the permanent magnet synchronous motor and employed 1DCNN to extract fault frequency characteristics for diagnosing rotor demagnetization [18]. Chan H. et al. utilized the Instantaneous Current Residual Map (ICRM) method to convert stator current signals of inter-turn short-circuit faults in permanent magnet synchronous motors into 2D images. Employing CNN for feature extraction and classification, they achieved a diagnostic accuracy of 94% [19], transforming one-dimensional vibration signals of rolling bearings into two-dimensional time-domain images and using a convolutional neural network (CNN) model to extract texture features facilitate fault diagnosis [20]. Another approach involves obtaining a one-dimensional vibration signal from the gearbox, utilizing a generative adversarial network (GAN) to extract and generate features of fractional fault sample data, followed by using a CNN for automatic feature extraction and fault classification [21]. Przemyslaw P. et al. proposed a method for detecting and classifying stator winding faults in PMSMs by combining bispectrum (BS) transformation with CNN applications. This approach transforms current signals into compressed bispectrum images, achieving a diagnostic accuracy of 99.4% [22]. Furthermore, incremental learning is widely utilized in fault diagnosis. Juan J. et al. introduced a method for multi-fault detection and identification based on incremental learning, achieving 99% accuracy by analyzing motor current features in electromechanical systems [23]. Yang B. et al. developed an online novelty detection scheme for motors using a one-class hyperdisk (OCHD) model, addressing underestimation issues associated with convex hull-based approaches, thereby effectively identifying motor faults during mass production and integrating incremental learning [24]. Zhao Z. et al. proposed an incremental learning (IL) method based on convolutional neural networks (CNNs), overcoming dependencies on large datasets and unknown classes, and achieving the efficient classification of unknown faults on wafer surfaces [25]. Compared to other feature extraction methods such as DBN [26], incremental learning [23,24,25], and various mathematical models [11,12,13], CNNs exhibit superior efficiency in extracting and learning useful features from raw data through their hierarchical structure of convolution and pooling operations and parameter-sharing mechanisms. This approach is particularly effective in processing complex image and time–frequency data. Consequently, this study has chosen to use CNNs for feature extraction. Furthermore, time–frequency analysis methods have been extensively employed in the field of fault diagnosis research, converting one-dimensional vibration signals from sources such as CWRU, gearbox gears, and motors into two-dimensional time-domain images [27,28]. These time–frequency methods primarily include CWT [22,29], STFT [30], DWT [26], HHT [15], and EMD [31]. Considering that CWT excels in detecting instantaneous changes and local features of signals, and demonstrates superior stability and robustness in noisy environments compared to EMD, the decision was made to utilize the continuous wavelet transform (CWT) for time–frequency analysis.

Despite significant contributions to fault diagnostics, these studies have several limitations, including exclusive reliance on mathematical or deep learning models, restriction to single fault types and severities, and limited diagnostic accuracy. Therefore, this study presents a mathematical model of the permanent magnet synchronous motor, analyzing the mechanisms of inter-turn short-circuit and demagnetization faults to identify key characteristic frequency ranges in the motor fault’s one-dimensional current signal. The continuous wavelet transform (CWT) method was employed to convert these signals into time–frequency images, which were used as model inputs. Convolutional neural networks (CNNs) were then utilized to extract deep image features for fault identification, with the training process visualized. This approach significantly improved fault diagnosis accuracy and revealed both local and global characteristics, enhancing interpretability. The main contributions are summarized as follows:This study, for the first time, combines mathematical modeling, mechanism analysis, and deep learning for the intelligent diagnosis of permanent magnet synchronous motor (PMSM) faults. It thoroughly analyzes inter-turn short-circuit and demagnetization faults and determines the key characteristic frequency ranges for different fault types, thereby reducing redundant feature interference.The fault diagnosis process is transformed into deep learning-based feature extraction and classification, with training visualization. By using CWT and STFT time–frequency analysis methods, one-dimensional current signals are converted into time–frequency images, and a CNN model is used for feature extraction and classification.A comprehensive fault diagnosis experiment is designed, covering 10 severity levels of inter-turn short-circuit and demagnetization faults, to verify the effectiveness of the proposed method.

The remaining sections of this work are organized as follows: Section 2 briefly introduces the mechanism of inter-turn short-circuit and demagnetization faults of permanent magnet synchronous motors and explains the basic principles of CWT and CNN. Section 3 provides detailed information on the proposed method. Section 4 outlines the workflow of the proposed method, including the dataset, selection of key characteristic frequency intervals, and data preprocessing. Experimental results are presented in Section 5 to demonstrate the effectiveness of the proposed method as shown by the confusion matrix and t-SNE. Finally, Section 6 provides a summary of the entire paper.

## 2. Basic Theory

### 2.1. Analysis of Inter-Turn Short-Circuit and Demagnetization Fault Mechanisms in PMSM

#### 2.1.1. Basic Structure and Operating Principles of PMSM

The permanent magnet synchronous motor primarily consists of a stator iron core, stator winding, rotor iron core, permanent magnet, and other components. Its structure is illustrated in Figure 1. Depending on the position of the permanent magnet in the rotor, the structure of the permanent magnet synchronous motor can be classified into two types: surface mounted and embedded. When the permanent magnet synchronous motor is at rest, three-phase symmetrical currents are applied to the stator winding to generate a stator rotating magnetic field and a rotor rotating magnetic field, thereby producing a starting torque for the motor. As the permanent magnet synchronous motor starts, the rotor winding ceases operation, and the motor is propelled by the magnetic field created by the permanent magnet and the stator winding.

Inter-turn short-circuit faults (ITFs) and demagnetization faults (DFs) of the stator winding are two common issues in permanent magnet synchronous motors. Inter-turn short-circuit faults may result from insulation layer aging due to mechanical, electrical, thermal, and other stresses, constituting 21% of all faults. Demagnetization faults of the permanent magnet may arise from high temperatures, chemical corrosion, strong magnetic fields, and control system malfunctions, comprising 26% of all faults [32].

#### 2.1.2. Analysis of Inter-Turn Short-Circuit Fault

It is assumed that the three-phase stator windings are connected in a star configuration, and the influence of the motor temperature rise on the resistance is disregarded. When an inter-turn short circuit occurs in the *A*-phase of the stator, the short-circuited section forms an additional loop, and the equivalent short-circuit resistance $R_{f}$ divides the *A*-phase winding $S_{a}$ into the normal segment and the short-circuited segment. The equivalent model of the motor’s three-phase stator winding inter-turn short-circuit fault is shown in Figure 2. $S_{b}$ and $S_{c}$ represent the two-phase windings of *B* and *C*, respectively. The three-phase current is denoted by $i_{a}$, $i_{b}$, and $i_{c}$.

When the permanent magnet synchronous motor experiences an inter-turn short-circuit fault, the harmonic frequency of the fault in the stator current can be expressed as follows:(1)$$f_{s}=\left(n\pm 2k\left(1-s\right)\right)f_{1},$$
where $f_{1}$ represents the fundamental frequency of the stator current; n=1,3,5,⋯; k=0,1,2,3,⋯; and *s* denotes the slip (close to 0). The amplitude of the third harmonic exhibits the most noticeable increase, and it is related to the severity of the inter-turn fault [33,34].

#### 2.1.3. Demagnetization Fault Analysis

When the motor experiences a demagnetization fault, the characteristic frequency of the fault in the stator current can be defined as: (2)$$f_{s}=\left(1\pm k/p\right)f_{1},$$
where $f_{1}$ represents the stator current fundamental frequency, with k=1,2,…; the stator current should exhibit 3/4 and 5/4 harmonic signals, which are real and serve as important features for fault diagnosis [35].

According to the mechanism analysis of stator inter-turn short-circuit faults and magnet demagnetization faults in permanent magnet synchronous motors, the frequency analysis interval is $\left[\frac{3}{4}f_{1},3f_{1}\right]$.

### 2.2. Basic Principle of Continuous Wavelet Transform (CWT)

The continuous wavelet transform method can adjust the size of its time window according to the signal frequency to extract high-frequency and low-frequency signals. It can also continuously amplify and translate the signal to ensure the accuracy of time–frequency analysis. It is suitable for processing non-stationary and non-linear signals. The mother wavelet [22] can be described by: (3)$$\psi_{m,n}\left(t\right)=\frac{1}{\sqrt{m}}\psi \left(\frac{t-n}{m}\right),$$
where $\psi_{m,n}$ is the basic wavelet (or mother wavelet), mainly including morlet, harr, morse, amor, and bump; the parameter *m* is the scaling factor (or scale factor) m>0, which controls the scaling of the wavelet basis function; and parameter *n* is the displacement factor, n∈R.

The continuous wavelet transform can be expressed as follows: (4)$$WT_{f}\left(m,n\right)=\langle f,\psi_{m,n}\rangle =\frac{1}{\sqrt{m}}\int f\left(t\right)\psi^{*}\left(\frac{t-n}{m}\right)dt,$$
where f(t) represents the signal. Due to the advantages of the continuous wavelet in dealing with non-stationary signals, CWT is used to convert the permanent magnet synchronous motor fault timing signal into images. In order to obtain higher time–frequency domain accuracy, the selected analytical wavelet is amor.

### 2.3. Basic Introduction to Convolutional Neural Network (CNN)

The convolutional neural network (CNN) is a type of feedforward neural network. In recent years, it has demonstrated outstanding performance in image recognition and object detection [36,37]. CNN is composed of multiple layers of neurons, and the neurons between layers are partially connected with weight sharing. The neurons in the same layer are independent of each other. In this paper, the input of CNN is a time–frequency image. After passing through convolutional layers, pooling layers, and fully connected layers, the output is the type of fault. The learning process of CNN is mainly divided into the forward propagation process and the backward parameter updating process.

#### 2.3.1. Forward Propagation Stage

The forward propagation stage of the convolutional neural network is mainly divided into three parts: the convolutional layer, pooling layer, and fully connected layer. In the convolutional layer, multiple convolution kernels are used to convolve the input image. After adding bias, a series of feature maps can be obtained through an activation function. The mathematical expression of the convolution process can be formulated as: (5)$$x_{j}^{L}=f\left(\sum_{i\in Y_{j}}x_{i}^{L-1}\cdot K_{ij}^{L}+b_{j}^{L}\right),$$
where $x_{j}^{L}$ is the *j* element of the *L* layer; $Y_{j}$ is the *j* convolution area of the L−1 feature map; $x_{i}^{L-1}$ is the element in it; $K_{ij}^{L}$ is the weight matrix of the corresponding convolution kernel; and $b_{j}^{L}$ is the bias term. f(•) is the activation function (the ReLU function is commonly used [38]), and the mathematical expression can be expressed as follows: (6)$$f\left(x\right)=\max\left(0,\mathrm{lg}\left(1+e^{x}\right)\right).$$

The pooling layer is often connected after the convolutional layer to reduce the dimensionality of the feature maps while maintaining a certain degree of invariance of the feature scale. Common pooling methods include Max Pooling, Mean Pooling, Stochastic Pooling, etc. The pooling layer generally only performs dimensionality reduction, has no parameters, and does not require updating weights. In the pooling layer, the feature map output by the convolutional layer is pooled in each non-overlapping area of size n×n, and the maximum or average value in each area is selected. Finally, the output image is divided into two regions, and the dimensions are all reduced by a factor of *n*. This process can be formulated as: (7)$$y^{c}=f\left(W^{c}X^{c-1}+B^{c}\right),$$
where *c* is the serial number of the network layer; $y^{c}$ is the output of the fully connected layer; $W^{c}$ is the weight coefficient; $X^{c-1}$ is the expanded one-dimensional feature vector; and $B^{c}$ is the bias term. f(•) is the activation function; Softmax was chosen as the activation function for the classification task.

#### 2.3.2. Backpropagation Weight Adjustment

The backpropagation (BP) process is used to adjust the network weights of each layer, minimizing the error cost function and reducing the difference between the predicted output and the actual result in order to establish a mapping space from the feature space to the fault space. CNN propagates the difference between the actual output *y* obtained through forward propagation and the expected output $\hat{y}$ obtained through backpropagation, and the BP algorithm adjusts the weights and biases between layers, layer by layer.

For a specific classification task, the training objective of CNN is to minimize the network’s loss function, so it is very important to choose an appropriate loss function. Common loss functions include Mean Squared Loss, Cross Entropy Loss, the Negative Log-likelihood (NLL) function, etc. The NLL function used in this article can be expressed as follows: (8)$$l\left(y_{i},{\hat{y}}_{i}\right)=-\log\hat{y}\left(k\right),$$
where $y_{i}$ is the *i* true label; ${\hat{y}}_{i}$ is the *i* predict label; and *k* represents the category.

## 3. The Proposed Method

The overall architecture of the fault diagnosis network model for permanent magnet synchronous motors, based on key frequency intervals, time–frequency analysis, and deep learning as proposed in this article, is illustrated in Figure 3. The model mainly consists of an input layer, four convolutional layers, four max-pooling layers, two fully connected layers, and an output layer.

The signals in the sample set undergo transformation into time–frequency images through continuous wavelet transform (CWT) to construct a feature image sample set. These time–frequency images are then further compressed to a size of 224 × 224 and transmitted to the input layer. The purpose of compression is to reduce the size of each dimension of the CNN input feature image, thereby improving the training speed of the network while ensuring that useful information in the time–frequency images is retained.

The convolutional layer primarily extracts time–frequency image features by computing the inner product of the two-dimensional convolution kernel and the corresponding input image’s overlapping area, iterating over each pixel on the entire image, and applying a non-linear activation function to obtain the output. The first and second convolutional layers have 32 and 64 convolution kernels, respectively, while the third and fourth convolutional layers have 128 convolution kernels. Each convolution kernel’s size is 3 × 3, and the ReLU function is chosen as the activation function. The pooling layer typically follows the convolutional layer to perform dimensionality reduction and parameter reduction operations on the output feature map of the convolution. Commonly used pooling layers include maximum pooling and average pooling. This article selects the maximum pooling layer and connects one after each convolutional layer. The size of the pooling layer is 3 × 3. The fully connected layer is employed for classification tasks. Alternatively, for regression tasks, a fully connected layer is utilized to further extract the output features of the pooling layer. For the classification task in this article, the Log_softmax activation function is commonly employed to normalize the output of the fully connected layer to conform to the probability distribution.

## 4. Dataset Description and Preprocessing

### 4.1. Current Signals for Inter-Turn Short-Circuit Fault and Demagnetization Fault

A real-time simulation platform based on hardware-in-the-loop for a permanent magnet synchronous motor was used to achieve real-time operational simulation of the permanent magnet synchronous motor drive system under normal conditions. The hardware components of the platform are shown in Figure 4, including the host computer, real-time simulator, control device, and motor. The collected data mainly consist of the measured data of the stator three-phase current sensor of the permanent magnet synchronous motor drive system.

Simulations were conducted by injecting demagnetization faults and inter-turn short-circuit faults to collect stator three-phase current sensor data of the permanent magnet synchronous motor drive system. Demagnetization faults and inter-turn short-circuit faults were injected to perform simulations and collect stator three-phase current sensor data of the permanent magnet synchronous motor drive system. The sampling frequency was set at 25 kHz, with two faults (demagnetization fault, inter-turn short-circuit fault), and 10 severity levels (10%, 20%, 30%, 40%, 50%, 60%, 70%, 80%, 90%, 100%), resulting in a total of 20 fault types. The sensor collected 20 seconds of data for each fault type and injected the fault at the fifth second, resulting in $2.35\times 10^{5}$ data points for each fault type. The data description is shown in Table 1.

### 4.2. Dataset Preprocessing

Time–frequency analysis techniques are widely used in the analysis and processing of non-stationary data. The current signals of demagnetization faults and turn-to-turn short-circuit faults are non-stationary one-dimensional signals whose local time–frequency characteristics are often of concern. Commonly used time–frequency analysis techniques include Short-Time Fourier Transform (STFT), continuous wavelet transform (CWT), Wigner–Ville Distribution (WVD), and Fourier Synchrosqueezed Transform (FST), among others. Additionally, convolutional neural networks (CNNs), known for their excellent recognition performance, have been widely used in fault diagnosis, improving both the accuracy and efficiency of fault detection. To convert a one-dimensional non-stationary signal into a two-dimensional numerical matrix that can be input into a CNN and to highlight the frequency domain characteristics of the one-dimensional time series signal, the current signal is processed through time–frequency transformation to enhance the effectiveness of CNN feature extraction. CWT can better describe the time–frequency local features of non-stationary signals in terms of time, frequency, and amplitude. It can also extract detailed features of signals by shifting or rescaling window functions. Therefore, CWT is chosen to convert the one-dimensional vibration signal into a time–frequency image.

According to the demagnetization failure mechanism, there should be 3/4 and 5/4 harmonic signals in the stator current when the motor is demagnetized. These harmonics are real and important features for fault diagnosis. Similarly, the inter-turn short-circuit failure mechanism predicts a $3f_{1}$ signal in the stator current when the motor suffers from an inter-turn short-circuit fault, with the amplitude of the $3f_{1}$ signal increasing as the fault severity increases. To verify this analysis, the current signals ITF30, ITF40, ITF70, ITF80, DF30, DF40, DF70, and DF80 were transformed into frequency spectra using fast Fourier transform (FFT) as shown in Figure 5. The time–frequency image obtained after continuous wavelet transform (CWT) is shown in Figure 6.

According to Figure 5a–d, the spectra of ITF30, ITF40, ITF70, and ITF80 contain second, third, and fourth harmonic frequencies. As the intensity of the inter-turn fault (ITF) increases, the amplitudes of different harmonics vary, but the amplitude variation of the third harmonic shows a good positive correlation with the fault severity. From Figure 5e–h, it can be seen that the harmonic frequencies of DF30, DF40, DF70, and DF80 are clearly present near the fundamental frequency, and there are always $3/4f_{1}$ and $5/4f_{1}$ harmonics present as the severity increases.

From Figure 6a–d, it is evident that when the one-dimensional current signal of the ITF is transformed into a time–frequency image, numerous non-key feature areas emerge in the upper part, while the main features are concentrated in the lower part, exhibiting values consistent with the harmonic frequencies of the fast Fourier transform. Transitioning to Figure 6e–h, a similar pattern emerges after transforming the one-dimensional current signal of the DF, with many non-key feature areas in the upper part and key features primarily concentrated in the lower part. Harmonics near the fundamental frequency are observable, with values consistent with the harmonic frequencies of the fast Fourier transform. Therefore, initially, the one-dimensional current data of both the ITF and DF undergo a fast Fourier transform to obtain a frequency spectrum, facilitating determination of the main harmonic frequency signal range. Subsequently, the continuous wavelet transform (CWT) is employed to transform the one-dimensional current signal within this main harmonic frequency range into a time–frequency image, thereby reducing the computing resources and accelerating the image conversion process.

Since both ITF and DF operate at a motor speed of 1800 rpm, their fundamental frequency is the same, denoted as $f_{1}$. Consequently, distinguishing between the two fault types ITF and DF, as well as assessing their severity, can be achieved by extracting key frequency intervals. This involves transforming the harmonic frequency intervals of ITF and DF, namely, $\left[3/4f_{1},3f_{1}\right]$, into time–frequency images, followed by feature extraction for fault diagnosis.

Based on spectral analysis of the rapidly transformed ITF and DF spectra, the key frequency range for the fundamental harmonic of ITF is $\left[f_{1},3f_{1}\right]$, while for DF, it is $\left[3/4f_{1},5/4f_{1}\right]$. Therefore, under identical experimental conditions, to encompass the main harmonic frequency range of ITF and DF, the time interval for fault diagnosis by transforming the one-dimensional current signals of ITF and DF into time–frequency images is set to [20Hz,400Hz]. The time–frequency images of key frequency intervals for ITF30, ITF40, ITF70, and ITF80 are depicted in Figure 7a–d, while those for DF30, DF40, DF70, and DF80 are shown in Figure 7e–h.

Analysis of Figure 7 reveals a significant reduction in non-critical feature areas of the ITF time–frequency image. Moreover, as severity increases, noticeable changes occur in the third harmonic area of the time–frequency image. Similarly, the non-critical feature areas of the DF time–frequency image undergo significant reduction, and with increasing severity, noticeable changes occur in the 3/4 and 5/4 harmonic areas of the time–frequency image. Consequently, the one-dimensional current signal within the ITF and DF frequency ranges of [0Hz,400Hz] is transformed into a time–frequency image using CWT and utilized as input for the CNN network.

## 5. Experiment Validation

### 5.1. PMSM Dataset Input

To verify the effectiveness of the proposed fault diagnosis method in this paper, a hardware-in-the-loop simulation platform based on a permanent magnet synchronous motor was utilized for experimental validation. The initial signals of the three-phase current data for stator winding inter-turn short-circuit faults and demagnetization faults were obtained. By analyzing the fault mechanism and the fast Fourier transform spectrum, the main characteristic frequency range for determining the fault type was identified as [20Hz,400Hz]. Therefore, the one-dimensional current signal of stator winding inter-turn short-circuit faults and demagnetization faults within the frequency range of [20Hz,400Hz] was transformed into a time–frequency image through CWT and inputted into a CNN network for fault diagnosis.

During the experiment, the batch size of the CNN model was set to 40, the epoch was set to 400, and the SGD optimization method was employed to update the parameters of the deep learning model through backpropagation, with a learning rate of 0.001. Please refer to Figure 3 for other network structure parameters. This dataset consists of 20 types of faults, divided into two categories, each with 10 severity levels as detailed in Table 1. Each fault type comprises 240,000 sample points, with each time–frequency image containing 5000 sample points and a 50% overlap. Hence, each fault type includes a total of 96 time–frequency images, with 66 designated for training and 28 for testing.

### 5.2. Experimental Results and Analysis

As depicted in Figure 8, the curves of training loss and test loss demonstrate consistent changes, as do the curves of training accuracy and test accuracy. For epochs less than 11, as the epoch increases, both training loss and test loss decrease rapidly, while training accuracy and test accuracy increase rapidly, with the test accuracy not exceeding 84%. For epochs greater than 36, the decrease in training loss and test loss slows down and stabilizes, while training accuracy and test accuracy slightly increase and converge. At this point, the training accuracy exceeds 96%, and the test accuracy exceeds 94%.

To enhance the interpretability and effectiveness of CNN networks, it is essential to understand how each layer transforms the input and to display the feature maps of intermediate layers in the network model. The 1st, 10th, and 20th channel feature maps of convolution layers (Conv 1, Conv 2, Conv 3, and Conv 4) and the feature maps extracted by the fully connected layers (FC 1 and FC 2) are displayed as shown in Figure 9, while the classification performance of each layer is depicted in Figure 10. Furthermore, to further illustrate the effectiveness of the proposed fault classification method for permanent magnet synchronous motors and the efficacy of training, T-SNE is utilized to map the high-dimensional and deep abstract features extracted by the trained model onto a two-dimensional plane as shown in Figure 11.

From Figure 9 and Figure 10, it is observable that Conv1 functions as a collection of various edge detectors, utilizing different convolution kernels across different channels to analyze the image from diverse perspectives. At this stage, the output post-activation almost retains all the information of the original image, containing numerous redundant features, thereby complicating the differentiation between various fault types. As the number of layers increases, the output becomes progressively more abstract, containing less information about the visual content of the image and more information about the fault types. Redundant features diminish, facilitating the differentiation between different fault types. Hence, it is evident that the CNN network model transforms the target into higher-level visual concepts, filtering out insignificant visual details to better distinguish between different categories. From Figure 11, it can be observed that as the number of epochs increases, the classification performance improves, further validating the effectiveness of the proposed method in this paper.

The confusion matrix serves as a tool for assessing the accuracy of a network model’s classification. To intuitively visualize the diagnostic effectiveness of the proposed method, the classification outcomes are summarized in a confusion matrix as depicted in Figure 12. In this matrix, the rows correspond to the predicted values generated by the method described in the article, while the columns correspond to the true values. The values along the diagonal indicate the number of accurate predictions, with higher values indicating better diagnostic performance.

### 5.3. Parameter Selection for the Proposed Method

To further validate the effectiveness of the proposed method, comparative experiments were conducted. The one-dimensional signal was transformed into a time–frequency graph using Short-Time Fourier Transform (STFT). Subsequently, these time–frequency graphs, without the selection of key feature frequency intervals, were input into the same network. Four datasets were utilized for these comparative experiments: KFICWT, CWT, KFISTFT, and STFT. In all experiments, the epoch was set to 100, with the other network structure parameters remaining consistent. The experimental test accuracies are illustrated in Figure 13.

Upon comparing the experimental results, it becomes evident that the time–frequency methods selecting key feature frequency intervals exhibit higher test accuracy. For instance, KFICWT achieves a classification accuracy of 98.6%, whereas CWT only reaches 50.1%. This discrepancy arises from the fact that selecting key feature frequency intervals reduces redundant features, enhances the feature extraction efficiency, and shortens the calculation time.

When comparing different time–frequency image conversion methods, the accuracy of KFICWT surpasses that of KFISTFT by 67.3%, reaching 98.6%. This difference can be attributed to the utilization of CWT of an adaptive time window, which enables better reflection of the frequency characteristics. Moreover, for the same STFT-based time–frequency analysis method, the accuracy of KFISTFT with selected key feature frequency intervals outperforms the STFT experimental results without such selection, further corroborating the effectiveness of this strategy.

In conclusion, the selection of key feature frequency intervals can reduce the interference of redundant features. Additionally, utilizing the CWT time–frequency analysis method can improve frequency feature resolution, thereby enhancing the accuracy of model classification. These results verify the effectiveness of the method proposed in this paper.

## 6. Conclusions

This paper proposes a CNN-based deep learning method for fault diagnosis of permanent magnet synchronous motors. The method is experimentally validated using a real-time simulation platform of a permanent magnet synchronous motor based on hardware-in-the-loop.

This study applies mathematical models and fault mechanism analysis in the intelligent diagnostics of permanent magnet synchronous motors (PMSMs) to investigate the underlying mechanisms of inter-turn short-circuit and demagnetization faults. The critical characteristic frequency range of these faults is identified as [20 Hz, 400 Hz]. To minimize redundant feature interference, the study focuses on current signals within this frequency range, converting them into time–frequency images using continuous wavelet transform (CWT) and Short-Time Fourier Transform (STFT). After evaluating both methods and considering the non-stationary nature of fault current signals, time–frequency images derived from CWT are selected as the model input. A convolutional neural network (CNN) model is developed to extract features from these images and perform classification, thus transforming the PMSM fault diagnosis process into one of deep learning-based feature extraction and classification, with detailed visualization of the training process.

This study includes a combined fault diagnosis experiment encompassing inter-turn short-circuit and demagnetization faults, each with ten severity levels, to validate the proposed method’s effectiveness. Experimental results demonstrate the method’s excellence in classifying these faults across various severity levels. The diagnostic accuracy of the KFICWT dataset reached 98.6%, significantly higher than the 50.1% of the CWT dataset. Additionally, the KFISTFT dataset achieved a diagnostic accuracy of over 67.3%, markedly higher than the 5% of the STFT dataset. Further analysis reveals that the KFICWT accuracy surpasses that of KFISTFT, indicating the superior effectiveness of CWT in representing fault characteristics. The method’s efficacy is further validated through t-SNE visualization and the clustering of features extracted by the CNN model layers.

However, this study has limitations, as it only analyzes current signals without considering other signals such as voltage and vibration. Additionally, it lacks experimental validation of PMSM faults using actual hardware and minor fault experiments. Future research will explore the diagnostic effectiveness of this method for slight PMSM faults (severity less than 5%) in practical experiments and consider multi-channel information fusion methods to enhance PMSM fault diagnostic accuracy.

## Figures and Tables

**Figure 1 sensors-24-06349-f001:**
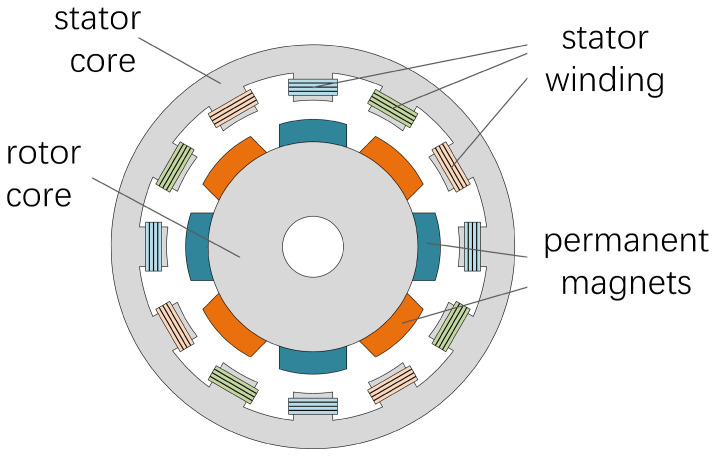
Structure of permanent magnet synchronous motor.

**Figure 2 sensors-24-06349-f002:**
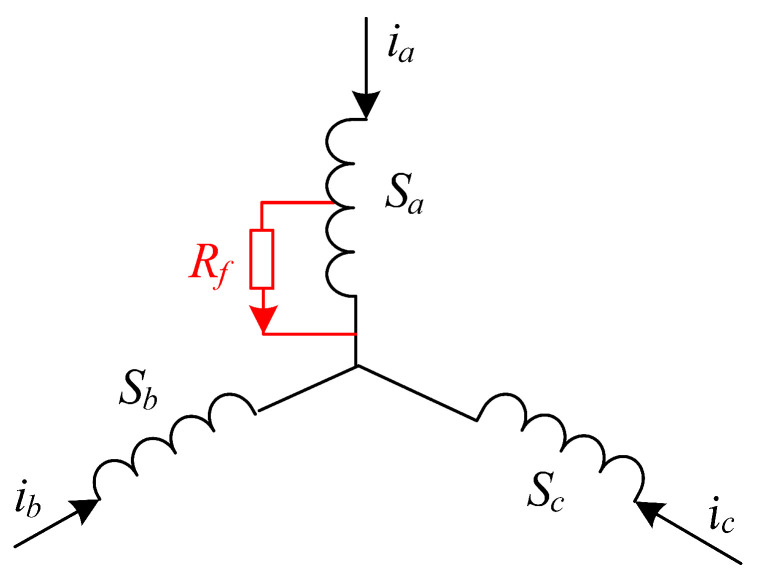
*A*-phase stator winding inter-turn short-circuit fault model.

**Figure 3 sensors-24-06349-f003:**
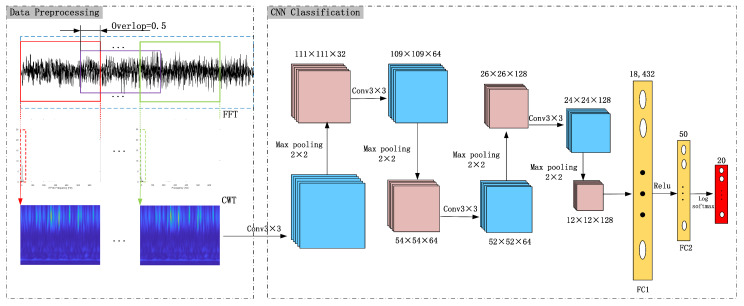
The framework of the proposed KFICWT-CNN model.

**Figure 4 sensors-24-06349-f004:**
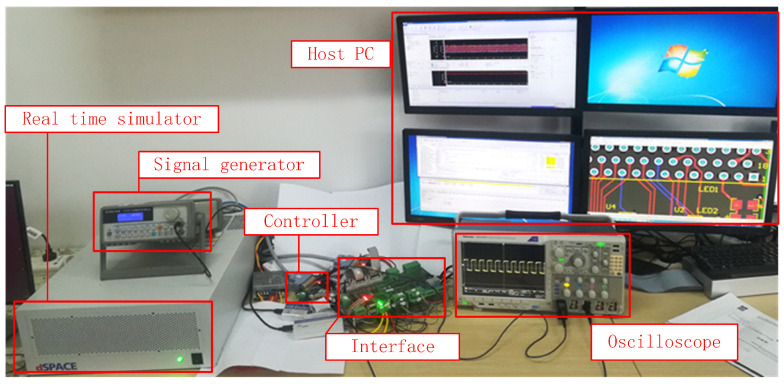
The hardware components of the platform.

**Figure 5 sensors-24-06349-f005:**
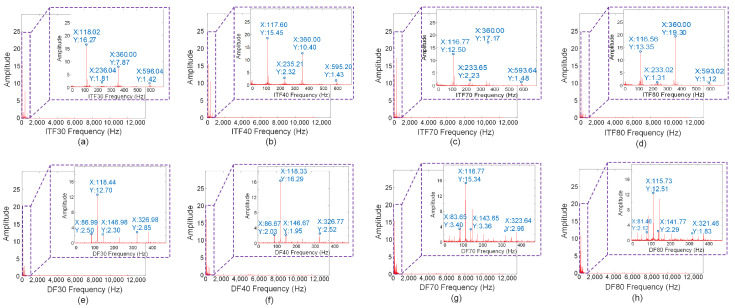
Fast Fourier transform spectra of ITF and DF.

**Figure 6 sensors-24-06349-f006:**
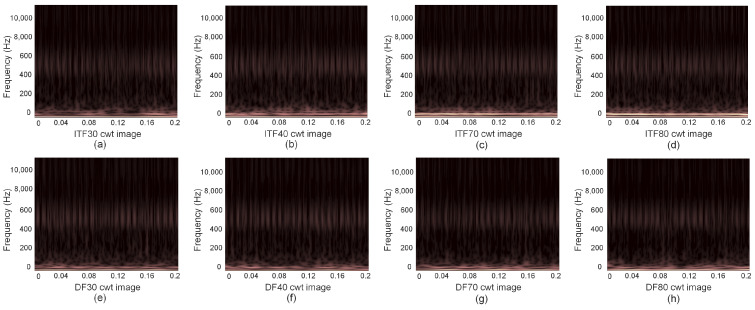
Continuous wavelet transform images of ITF and DF.

**Figure 7 sensors-24-06349-f007:**
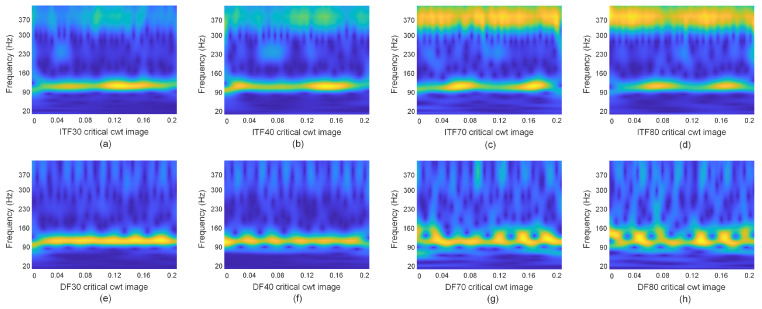
The key frequency interval time–frequency images of ITF and DF.

**Figure 8 sensors-24-06349-f008:**
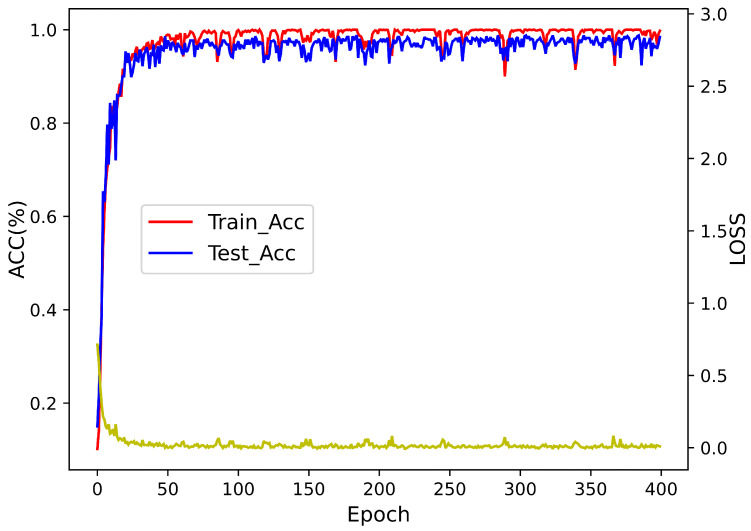
Loss and accuracy of KFICWT-CNN.

**Figure 9 sensors-24-06349-f009:**
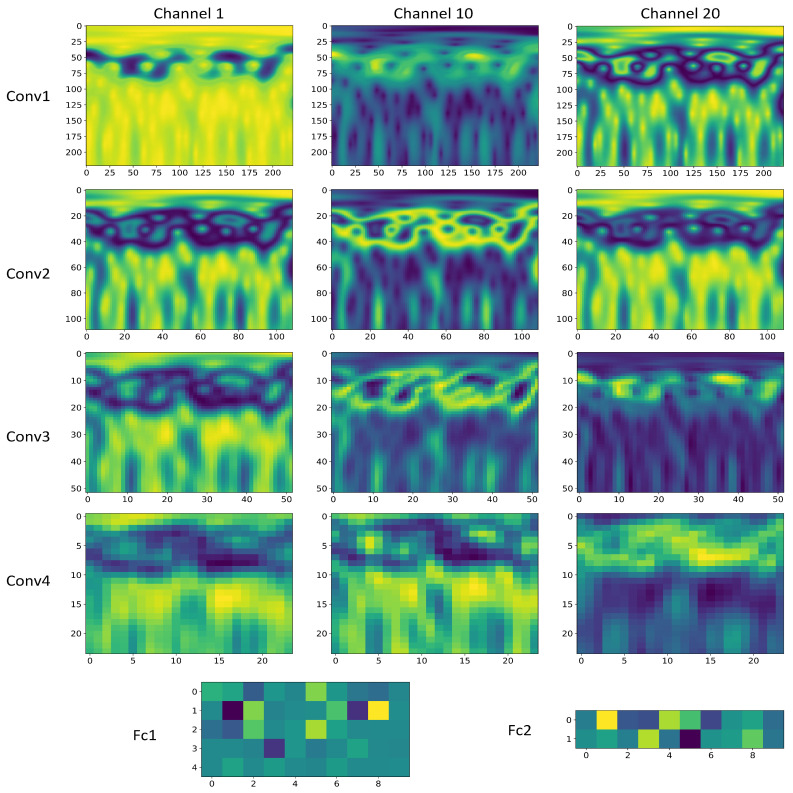
Different channels feature of KFICWT-CNN.

**Figure 10 sensors-24-06349-f010:**
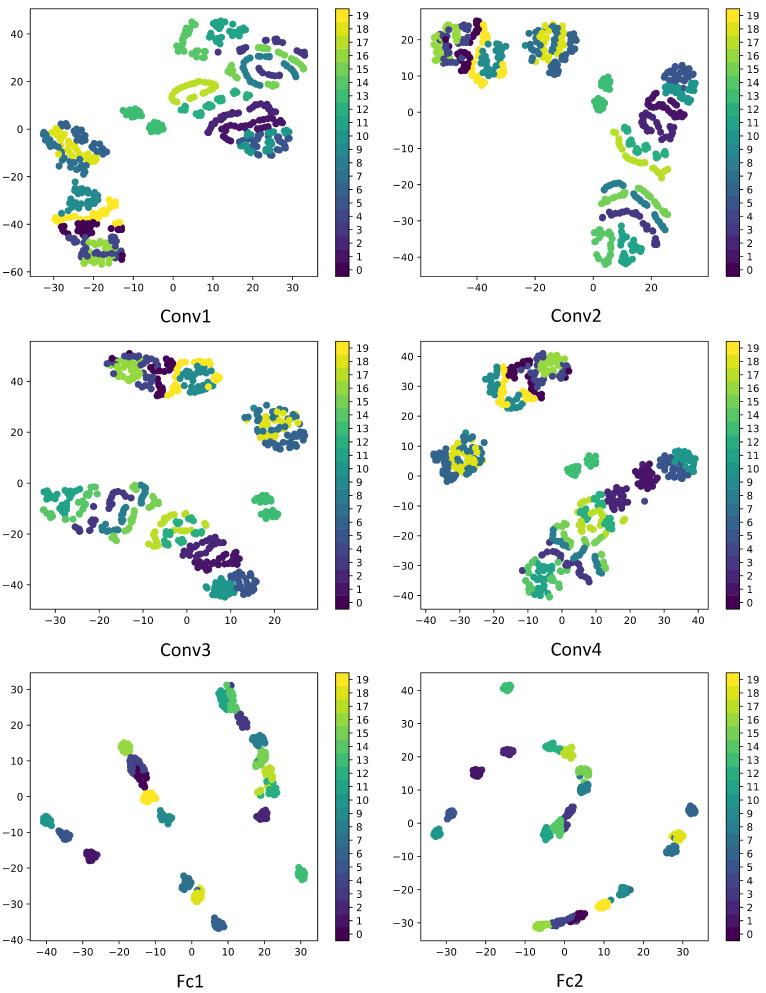
T-SNE classification results of feature maps from different layers.

**Figure 11 sensors-24-06349-f011:**
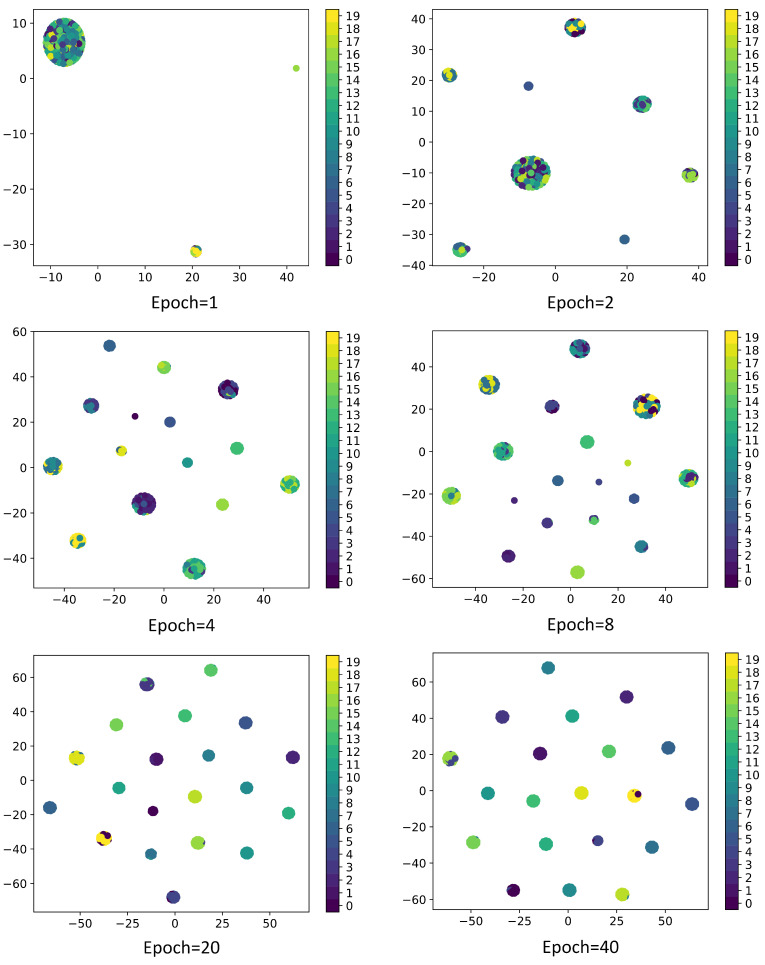
Different epoch classification results of KFICWT-CNN.

**Figure 12 sensors-24-06349-f012:**
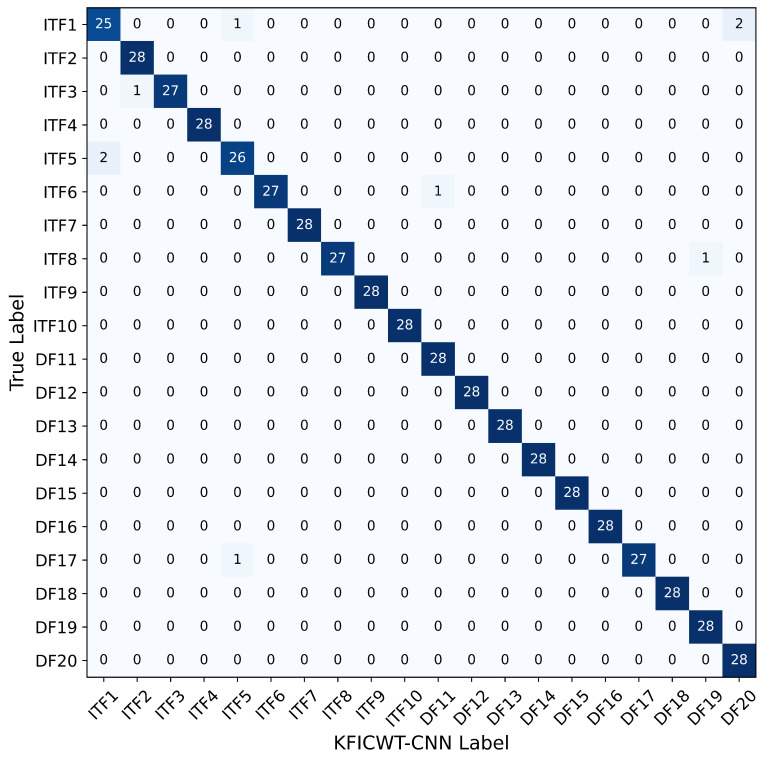
KFICWT-CNN of confusion matrix.

**Figure 13 sensors-24-06349-f013:**
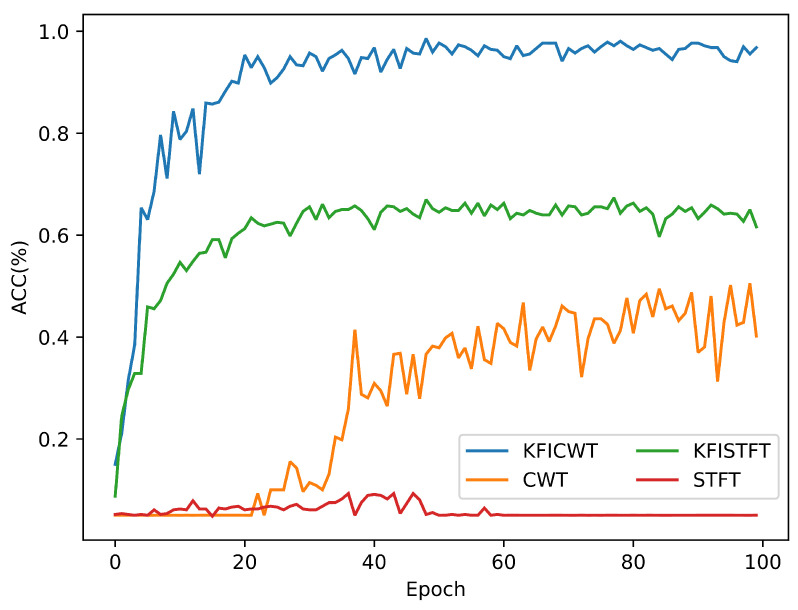
Testing accuracy of comparative experiments.

**Table 1 sensors-24-06349-t001:** Permanent magnet synchronous motor demagnetization fault and inter-turn short-circuit fault types.

Fault Type	Fault Levels (%)	Motor Speed (rpm)	Label
Inter-turn short fault	10/20/30/40/50/ 60/70/80/90/100	1800	ITF1∼ITF10
Demagnetization fault	10/20/30/40/50/ 60/70/80/90/100	1800	DF11∼DF20

## Data Availability

Some or all data, models, or code that support the findings of this study are available from the corresponding author upon reasonable request.

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
