# Peer review of "Mechanism-Based Fault Diagnosis Deep Learning Method for Permanent Magnet Synchronous Motor"

_sensors, 2024, doi:10.3390/s24196349_

Round 1

Reviewer 1 Report

Comments and Suggestions for Authors

I recommend that the authors take into consideration the following comments:

1. Explain the main contribution that this manuscript brings in comparison to the current state of the art.

2. Provide a detailed explanation of the specific improvements that were made to the approach.

3. Discuss the contribution of the research by comparing it to related work, if there any.

Author Response

Please see the attachment.(Reviewer_1.pdf)

Reviewer 2 Report

Comments and Suggestions for Authors

In this work is proposed a diagnosis strategy based on Wavelet transform and Convolutional Neural Network for identifying different condition related to inter-turn short circuit in a permanent magnet synchronous motor.

The proposal is interesting but there are some issues that must be addressed.

1. Please avoid to write in first person, i.e., replace phrases like "... we propose an intelligent fault diagnosis method based on continuous wavelet transform (CWT...)"  with  "... in this work is proposed an intelligent fault diagnosis...". This comment applies for all section of the whole manuscript.

2. The novelty and contribution need to be highlighted in order to identify the most important differences with other previous published works.

3. The Introduction section is composed of several small paragraphs, I suggest to reorganize this section with the aim of improve the readability. Also, there are some previously works that has been published to identify faults in PMSM, some of them are important since have propose diagnosis structures that lead to efficient results, please consider to include a brief discussion of the following papers in the Introduction:  https://doi.org/10.1109/TIE.2022.3189076 ;  https://doi.org/10.1109/TII.2020.2973731 ; 10.1088/1361-6501/ad490e

4. I suggest to separate the description of the Experimental Setup and the description of the Proposed method, also, I Suggest first include the flow chart of the proposed method and then include the results.

5. What are the main advantages of using CWT in front of other time-frequency techniques like EMD?

6. What are the limitations of this proposal? include a brief description in the Conclusion section.

Reviewer 3 Report

Comments and Suggestions for Authors

This paper proposed a mechanism-based fault diagnosis deep learning method for permanent magnet synchronous motor. In general, this paper is well written, and my comments are given as follows:

1. When a formula is part of a sentence in the paper, do not use a colon after "is"; use a colon when it follows "as follows." The formula should have the appropriate punctuation at the end to maintain sentence integrity and readability. Please have the author carefully review the entire text.   2. The innovation of the article is not obvious, and the time-frequency conversion and feature extraction methods used have not been improved before use.   3. The data in the article was obtained on a simulation platform, and there is a lack of experimental data to verify the effectiveness of the proposed method.   4. The proportion of figures 4 to 6 on the page of the article is too large, and the author should improve the layout. Additionally, there is an error in ITF30 in Figure 4b.   5. The color contrast in Figure 5 is not obvious, and the author should replace it with a more contrasting color to improve the readability of the article.   6. The article mainly compares and analyzes the results in time-frequency conversion, and there is not much theoretical analysis on how to choose. In addition, there is a lack of comparison in feature extraction.   7. Other related fault diagnosis methods are suggested to be discussed in the literature review, e.g., incremental learning, An Unknown Wafer Surface Defect Detection Approach Based on Incremental Learning For Reliability Analysis, Reliability Engineering & System Safety 244 (3), 109966. Comments on the Quality of English Language

Minor editing of English language is required.

Round 2

Reviewer 2 Report

Comments and Suggestions for Authors

The authors have addressed all the comments and the quality of the manuscript has increased, the proposed work can be accepted in its current form after editor decision.

Reviewer 3 Report

Comments and Suggestions for Authors

The authors have addressed all my comments.

Comments on the Quality of English Language

Minor editing of English language is required.